# Treatment of Equine Tarsus Long Medial Collateral Ligament Desmitis with Allogenic Synovial Membrane Mesenchymal Stem/Stromal Cells Enhanced by Umbilical Cord Mesenchymal Stem/Stromal Cell-Derived Conditioned Medium: Proof of Concept

**DOI:** 10.3390/ani14030370

**Published:** 2024-01-24

**Authors:** Inês Leal Reis, Bruna Lopes, Patrícia Sousa, Ana Catarina Sousa, Mariana V. Branquinho, Ana Rita Caseiro, Alexandra Rêma, Inês Briote, Carla M. Mendonça, Jorge Miguel Santos, Luís M. Atayde, Rui D. Alvites, Ana Colette Maurício

**Affiliations:** 1Departamento de Clínicas Veterinárias, Instituto de Ciências Biomédicas de Abel Salazar (ICBAS), Universidade do Porto (UP), Rua de Jorge Viterbo Ferreira, n° 228, 4050-313 Porto, Portugal; lealreines@gmail.com (I.L.R.); brunisabel95@gmail.com (B.L.); pfrfs_10@hotmail.com (P.S.); anacatarinasoaressousa@hotmail.com (A.C.S.); m.esteves.vieira@gmail.com (M.V.B.); alexandra.rema@gmail.com (A.R.); ines_briote@hotmail.com (I.B.); cmmendonca@icbas.up.pt (C.M.M.); jmposs1970@gmail.com (J.M.S.); ataydelm@gmail.com (L.M.A.); ruialvites@hotmail.com (R.D.A.); 2Centro de Estudos de Ciência Animal (CECA), Instituto de Ciências, Tecnologias e Agroambiente da Universidade do Porto (ICETA), Rua D. Manuel II, Apartado 55142, 4051-401 Porto, Portugal; rita.caseiro.santos@gmail.com; 3Associate Laboratory for Animal and Veterinary Science (AL4AnimalS), 1300-477 Lisboa, Portugal; 4Cooperativa de Ensino Superior Politécnico e Universitário (CESPU), Avenida Central de Gandra 1317, 4585-116 Gandra, Portugal; 5Departamento de Ciências Veterinárias, Escola Universitária Vasco da Gama (EUVG), Avenida José R. Sousa Fernandes, Lordemão, 3020-210 Coimbra, Portugal; 6Centro de Investigação Vasco da Gama (CIVG), Escola Universitária Vasco da Gama (EUVG), Avenida José R. Sousa Fernandes, Lordemão, 3020-210 Coimbra, Portugal; 7Campus Agrário de Vairão, Centro Clínico de Equinos de Vairão (CCEV), Rua da Braziela n° 100, 4485-144 Vairão, Portugal

**Keywords:** allogenic, conditioned medium, equine, ligament, mesenchymal stem/stromal cells, MSC-based therapies, orthobiologic, secretome, synovial membrane, umbilical cord

## Abstract

**Simple Summary:**

Musculoskeletal injuries in sport horses are relatively common and quite worrisome. Tendon and ligament injuries in sport horses usually result in a long period of time out of competition. Their healing usually results in tissue fibrosis and concomitant loss of elasticity, which, depending on the severity, might prevent the horse’s recovery to the same performance levels or even to athletic levels. The continuous development of regenerative medicine offers therapeutical promise. Synovial membrane mesenchymal stem/stromal cells (SM-MSCs) and umbilical cord mesenchymal stem/stromal cells (UC-MSC), as well as their growth factors, have been described as having optimal characteristics for tendon and ligament regeneration. Therefore, a therapeutical combination of SM-MSC and a conditioned medium of UC-MSC was developed, produced, and administered on a tarsal long medial collateral ligament desmitis of a show-jumping horse. The production and application of the orthobiologic therapeutical combination as well as the clinical outcome are presented herein.

**Abstract:**

Horses are high-performance athletes prone to sportive injuries such as tendonitis and desmitis. The formation of fibrous tissue in tendon repair remains a challenge to overcome. This impels regenerative medicine to develop innovative therapies that enhance regeneration, retrieving original tissue properties. Multipotent Mesenchymal Stem/Stromal Cells (MSCs) have been successfully used to develop therapeutic products, as they secrete a variety of bioactive molecules that play a pivotal role in tissue regeneration. These factors are released in culture media for producing a conditioned medium (CM). The aforementioned assumptions led to the formulation of equine synovial membrane MSCs (eSM-MSCs)—the cellular pool that naturally regenerates joint tissue—combined with a medium enriched in immunomodulatory factors (among other bioactive factors) produced by umbilical cord stroma-derived MSCs (eUC-MSCs) that naturally contribute to suppressing the immune rejection in the maternal–fetal barrier. A description of an equine sport horse diagnosed with acute tarsocrural desmitis and treated with this formulation is presented. Ultrasonographic ligament recovery occurred in a reduced time frame, reducing stoppage time and allowing for the horse’s return to unrestricted competition after the completion of a physical rehabilitation program. This study focused on the description of the therapeutic formulation and potential in an equine desmitis treatment using the cells themselves and their secretomes.

## 1. Introduction

Tendon and ligament injuries account for a large proportion of a sport horse’s wastage and early retirement. Most injuries are overstrained injuries, as these structures operate near their functional limits in sport horses. Tendons and ligaments are highly organized tissues that depend on the strength and structure of the extracellular matrix to function [1]. Overloading can lead to physical damage and degeneration [1,2]. Although tendons and ligaments can heal spontaneously with time, the fibrous scar tissue formed is biomechanically inferior, leading to a decrease in tissue functionality and, therefore, in athletic performance, as well as recurrent reinjuries and lameness. Tarsocrural desmitis is an inflammatory event, causing moderate to severe hindlimb lameness in horses, and it is often clinically underdiagnosed due to unspecific clinical signs such as synovial effusion and minimal radiographic changes [3]. Ultrasonographic examination allows for desmitis diagnosis characterized by increased ligament size, decreased echogenicity, and abnormal fiber pattern. This pathology has a guarded prognosis for athletic soundness [4]. 

Medial tarsal collateral desmitis has been rarely described in the veterinary literature; however, it is one of the most common soft tissue injuries of the tarsus. Conventional treatments are often referred to in the literature, but the outcome is not very advantageous, since there is a guarded prognosis to return to the same performance level, and osteoarthritis (OA) is usually secondarily associated with these approaches [4,5,6,7,8,9]. 

In an attempt to overcome these obstacles, regenerative therapies are gaining increased interest, and the use of multipotent Mesenchymal Stem/Stromal Cells (MSCs) therapy holds immense promise [4,5,6]. The MSCs are adult multipotent progenitor cells found in many organs and tissues, able to self-renew, to migrate to injury sites (homing), to differentiate into multiple lineages, and to secrete bioactive factors, providing immunomodulation, increasing proliferation and migration of tendon stem/progenitor cells via paracrine signaling, and increasing the regeneration ability of tissues with a poor aptitude [10,11,12,13]. 

Interestingly, in light of the many findings regarding their identity and function, the adequate nomenclature of the MSCs acronym has become itself a field of debate. Classically, the “Mesenchymal Stem Cells” designation, as proposed by the International Society for Cellular Therapy (ISCT) in defining the minimal criteria, has been the most widely accepted [14]. However, other scientific voices currently advocate that these do not represent “true” stem cells, since there is still a lack of indisputable evidence of some critical stemness features (such as the asymmetric cell division and complete lineage renewal). Thus, the term “Mesenchymal Stromal Cell” is proposed to better describe the conventionally isolated populations [15]. Recently, it was proposed to rename it to “Medicinal Signaling Cells” to more accurately reflect the fact that these cells home in on sites of injury or disease and secrete bioactive factors that are immunomodulatory and trophic (regenerative), meaning that these cells make therapeutic drugs in situ, which are medicinal [16]. Therefore, the aim of regenerative stem cell medicine is to regenerate cells and tissues and to restore their normal structure and function [1,3]. These therapies consist in the administration of MSCs themselves or MSC free products [1,17,18].

The latest studies highlight the importance of the paracrine action of MSCs through the release of soluble and non-soluble factors, primarily secreted in the extracellular space by MSCs, known as the secretome [19] whose paracrine signaling is considered the primary mechanism by which MSCs contribute to healing processes [20,21], avoiding the need for living-cell implantation [22,23]. This spurs the design of MSC-based therapies that do not require cell administration being immediately available for the treatment of acute conditions, with the possibility of being massively produced from commercially available cell lines, avoiding invasive cell collection procedures [11,19,24,25]. Stem-cell-free products have demonstrated preclinical efficacy and safety, as they appear to have non-cytotoxicity, non-mutagenicity, and low immunogenicity. They also overcome the challenge of cell viability maintenance and potency throughout the manufacture, storage, and delivery, maintaining the advantages of therapeutic ability [26]. Several cell-free preparations have shown encouraging outcomes in early-stage clinical trials [27].

Equine synovial membrane and umbilical cord mesenchymal stem cells (SM-MSCs and UC-MSCs), the object of our research, are an interesting alternative cellular and cell-based therapy due to their promising articular, tendon, and ligamentous high regenerative capacity. Additionally, the evaluation of these cells’ secretome is important to understand their biological potential and their synergistic action, as MSCs isolated display similar appearances but different biological functions and markers depending on the their origins [27]. Equine SM-MSCs’ tenogenic superiority presents them as good candidates for tendon and ligament regeneration [28,29,30]. Also, it has been proven that UC-MSCs present a superior ability in differentiating into tendon-like lineages and forming a tendon-like matrix, improving the regeneration of these structures [31]. Additionally, the UC-MSC secretome is characterized by the presence of angiogenic factors, reduced levels of metalloproteinases (MMPs), and elevated synthesis of transforming growth factor β 1 (TGF-β1), chemokines, and anti-inflammatory cytokines, with interleukyn-6 (IL-6) being one of the most secreted. These characteristics suggest that the UC-MSC secretome presents the ability to control inflammatory responses [32,33]. 

The aim of the present research is to evaluate the beneficial effect of a newly developed stem-cell orthobiologic therapeutic combination, consisting of eSM-MSCs and a UC-MSC conditioned medium (CM), in the treatment of equine tarsal ligament desmitis. Herein, a diagnosis and innovative treatment approach of a seven-year-old show-jumping horse, who sustained an injury of the long medial collateral ligament (LMCL) of the right tarsus, is thoroughly described. Additionally, the orthobiologic product preparation and application, as well as the determination of the eSM-MSC and UC-MSC therapeutic potential, using the cells themselves and the cell-derived CM, are demonstrated. 

## 2. Materials and Methods

### 2.1. Ethics and Regulation

This research was carried out in accordance with “Organismo Responsável pelo Bem Estar Animal” (ORBEA) from ICBAS-UP recommendations and authorization (reference P289/ORBEA/2018). Treatments were performed with permission and signature of informed consent from the patient’s legal tutor after a thorough explanation of the procedure itself and possible associated risks, in accordance with the national legislation from the competent authorities.

### 2.2. Patient Identification and Clinical Evaluation

A seven-year-old showjumper stallion was examined for a complaint of a swollen right tarsus. Upon examination, signs compatible with acute lesion of the right tarsus, such as oedema and significant effusion of the tarsocrural joint, were observed (Figure 1). 

The animal was subjected to a complete physical and orthopedic examination including dynamic examination, where the evaluation of lameness was performed while walking, trotting, and galloping in a straight line and in circles, on both rough and soft surfaces. Flexion tests of the main joints, followed by trotting in a straight line on a rough surface, were also performed, and lameness was scored on a scale proposed by the American Association of Equine Practitioners (AAEP) (Table 1).

On the first assessment day, complementary diagnostic examinations included radiographs and ultrasound images, as reported in other studies [5,6,7]. The follow-up evaluations of the patient included a physical examination, as well as lameness and ultrasonographic evaluations. 

### 2.3. Complementary Diagnostic Exams

#### 2.3.1. Radiological Examination

Radiological examination (X-ray) of the right tarsocrural joint was performed with a digital system—CareRay Cw series^®^ (CareRay, Suzhou, China), radiological constants: 72 Kv, 0.8 mA. The distance between the X-ray generator (Orange 1060 HF, EcoRay, Seoul, Republic of Korea) and the flat panel was approximately 66 cm. Four standard views—lateromedial, dorsoplantar, dorsolateral-plantaromedial and dorsomedial-plantarolateral—were obtained at patient’s first examination.

#### 2.3.2. Ultrasound Examination

Ultrasound examination (U/S) of the right and left tarsocrural joint was performed in longitudinal and transversal scans using a 7.5-MHz linear probe, digital portable ultrasound system—Sonoscape A6^®^ (SonoScape, Shenzhen, China).

The contralateral limb was considered normal and used as control. Echogenicity, fiber pattern, and cross-sectional area were evaluated in each collateral ligament. The synovial fluid was evaluated for signs of hemarthrosis (increase in echogenicity and/or a swirling echogenic pattern). The synovial lining was evaluated for thickening and fibrinous loculations in the tarsocrural joint. The medial and lateral long and short collateral ligaments of the tarsus were examined in longitudinal and transverse planes, from proximal to distal. 

### 2.4. Donor Selection and SM Collection

The equine SM-MSCs’ donor was a young and healthy foal (six months’ old) who suffered an accident and died. Briefly, the tutor authorized synovial membrane collection from hocks, knees, and fetlocks. Before the procedure, the incisional field was subjected to surgical asepsis with chlorohexidine and alcohol. Sequentially, the skin was incised, the subcutaneous tissue was debrided, the joint capsule was incised and opened, and the synovial membrane was isolated and extracted. The collected tissues were immersed in Dulbecco’s phosphate-buffered saline (DPBS—14190-144, Gibco^®^, Waltham, MA, USA) and transported to the laboratory facility, wrapped in ice packs to maintain refrigeration. These procedures have been previously described [28].

### 2.5. Equine SM-MSCs’ Isolation, Culture, and Characterization

The collected synovial tissue was processed at the Laboratory of Veterinary Cell-based Therapies—ICBAS-UP, following the patented eSM-MSCs’ isolation protocol (PCT/IB2019/052006, WO2019175773—“Compositions for use in the treatment of musculoskeletal conditions and methods for producing the same leveraging the synergistic activity of two different types of mesenchymal stromal/stem cells”—Regenera^®^, Barcelona, Spain) previously described [28].

Before cryopreservation of the cells’ batch, the culture medium was subjected to a bacteriological control using the BACT/ALERT^®^ (BioMérieux Portugal^®^, Linda-a-Velha, Portugal) medium to rule out the presence of any bacterial or fungal contamination.

### 2.6. Equine UC-MSCs’ Isolation, Culture, and Characterization

Equine UC-MSCs were isolated from the equine umbilical cord matrix (Wharton’s jelly), a birth residue, collected during a full-term parturition of a different foal from which the synovial tissues were obtained.

Briefly, tissue samples were collected and placed in transport media supplemented with penicillin (300 U/mL)–streptomycin (300 μg/mL) (15140-122—Gibco^®^) and amphotericin B (5 μg/mL) (15290-026—Gibco^®^). Upon arrival, umbilical cord tissues were decontaminated and dissected for the isolation of the stromal tissue, which was digested using Colagenase I (17100-017—Gibco^®^) and Dispase II (17105-041—Gibco^®^). Single-cell suspension of the digested tissues was obtained through a 70 µm cell strainer (CLS431751—Corning^®^ Falcon^®^, Corning, NY, USA) and cultured in DMEM-HG (11965092—Gibco^®^), 20% (*v*/*v*) MSC qualified FBS (04-400-1A, Biological Industries Israel Beit-Haemek Ltd., Migdal HaEmek, Israel), penicillin (150 U/mL)–streptomycin (150 μg/mL), and amphotericin B (3.75 μg/mL) for the first 24 h. Non-adherent cells were discarded after 24 h, and the remaining cells were further expanded in DMEM-LG (11885-084—Gibco^®^), 10% (*v*/*v*) FBS, penicillin (100 U/mL)–streptomycin (100 μg/mL) (Gibco^®^), and amphotericin B (2.5 μg/mL) (Gibco^®^) to form a culture of adherent cells with fibroblastic morphology. This process was performed and is patented proprietary technology (PCT/IB2019/052006, WO2019175773—Regenera^®^).

Before cryopreservation of the cells’ batch, the culture medium was subjected to a bacteriological control using the BACT/ALERT^®^ (BioMérieux Portugal^®^) medium to rule out the presence of any bacterial or fungal contamination.

As previously described, isolated populations’ immunophenotypes were validated in Passage 5 [28]. The antibodies were selected to confirm the mesenchymal/stromal histogenesis and discard epithelial and endothelial histogenesis of eUC-MSCs. For each antibody, appropriate negative and positive controls were included, and all were incubated overnight. Detailed immunocytochemistry information is provided in Appendix A.

Trilineage differentiation was also validated in Passage 5: adipogenic, chondrogenic, and osteogenic differentiation (StemPro^®^ Differentiation kits: A1007201—Osteogenesis; A1007001—Adipogenesis; A1007101—Chondrogenesis, Gibco^®^) were assessed using Oil Red-O, Alcian Blue, and Von Kossa stainings, respectively. All protocols were performed as described [28], except for Von Kossa stain employed to detect mineral extracellular deposition, in which cells were fixated and dehydrated with increasing ethanol concentrations and then rehydrated and incubated in 2% Silver Nitrate (7761-88-8, Sigma-Aldirch, St. Louis, MO, USA) solution under UV light and sodium thiosulfate (1091471000, Merk^TM^, Rahway, NJ, USA) 5% for 3 min. Wells were rinsed, and microphotographic records were obtained.

### 2.7. Secretome—Conditioned Medium Preparation and Analysis

The conditioned medium of eSM-MSCs and eUC-MSCs in Passages 4 and 6, respectively, was analyzed to identify cytokines and chemokines secreted after conditioning. When in culture, after reaching a confluence of around 70–80%, the culture medium was removed, and the culture flasks were gently washed with DPBS two to three times. Then, the culture flasks were further washed two to three times with the basal culture medium of each cell type without any supplementation. The conditioning procedure was based on maintaining the culture flasks in standard culture conditions after the addition of non-supplemented DMEM/F12 GlutaMAX™ (31331-093—Gibco^®^). The culture flasks were kept under these conditions for 48 h, stimulating the cells to secrete paracrine factors (CM). After this period, the CM was collected and subjected to centrifugation at 3000× *g* for 10 min. The sediment was eliminated by collecting the supernatant, which was filtered with a 0.2 μm syringe filter (Filtropur S, PES, Sarstedt^®^, Nümbrecht, Germany). Then, the CM was concentrated five times (5×) using a Pierce™ Protein Concentrator, 3k MWCO, 5–20 mL tubes (88525, Thermo Scientific^®^, Waltham, MA, USA). The concentrators were sterilized before the procedure. Initially, the upper compartment of each concentrator tube was filled with 70% ethanol (*v*/*v*) and centrifuged for 10 min at 3000× *g*. Then, the ethanol was discarded, and the same procedure was carried out with DPBS. Each concentrator tube was subjected to two such centrifugation cycles, followed by a 10 min period in the laminar flow hood for complete drying. Finally, the upper compartment of the concentrator tubes was filled with plain CM (1× concentration) and subjected to a new centrifugation cycle, under the conditions described above, for the number of cycles necessary to obtain the desired CM concentration (5×).

Equine SM-MSCs secretome has been previously characterized [28]. Similarly, eUC-MSCs CM was stored at −20 °C, concentrated 5×, and assessed for the production and secretion of the selected equine biomarkers, IL-6 and IL-8, through Multiplexing LASER Bead analysis (Eve Technologies, Calgary, AB, Canada). The average concentration for each interleukin in eUC-MSC was evaluated in triplicate.

### 2.8. Equine SM-MSCs + eUC-MSC CM Solution Preparation

The eSM-MSCs solution for intra-ligamental clinical application was a combination of allogeneic eSM-MSCs suspended in eUC-MSCs CM.

Cryopreserved P3 eSM-MSCs batches were thawed and suspended in the recipient animal’s heat-inactivated autologous serum. For this purpose, 10 mL of whole blood was collected into two serum tubes (367820—BD Vacutainer^®^, Franklin Lakes, NJ, USA). The tubes were then centrifuged at 6000× *g* for 10 min, and the supernatant (autologous serum) was collected into a 15 mL tube. The serum sample was heat-inactivated for 20 min at 56 °C (water bath), quickly cooled down in an ice bath, and sterile-filtered with a syringe filter with 0.22 µm into a new 15 mL tube. For one 9 × 10^6^ eSM-MSCs dose, 3 × 2 mL eSM-MSCs vials containing around 3 × 10^6^ cells each were thawed in a 37 °C water bath, and the cell suspensions of the 3 vials were mixed into one 15 mL tube. Then, 2–3 mL of autologous serum was slowly added to the tube (drop-wise), and the suspension was gently mixed. A total of 5 mL of PBS was slowly added into the tube, and the suspension was gently mixed and centrifuged at 3000× *g* for 10 min. After eliminating the supernatant, the cell pellet was resuspended in autologous serum, maintaining a ratio of 0.8:1. The cells were then counted, and their viability was determined in an automatic counter (Countess II FL Automated Cell Counter, Thermo Fisher Scientific^®^, Waltham, MA, USA) using the Trypan Blue exclusion dye assay (T10282—Invitrogen^TM^, Waltham, MA, USA). Around 8.58 × 10^6^ cells were counted with an average viability of 98% (triplicate count) before the first administration, and 9.42 × 10^6^ cells were counted with an average viability of 100% (triplicate count) before the second one. The cell number was then adjusted to 1 × 10^7^ cells/mL. At this point, the conditioned medium from eUC-MSCs was thawed and added to the suspension to a final 1:1 concentration. Then, 2 mL of the solution of eSM-MSCs suspended in eUC-MSCs CM was transferred to a perforable capped vial and preserved on ice until the moment of administration.

### 2.9. Treatment Protocol

The injured structure—LMCL—was treated with the mixture of allogenic eSM-MSCs and eUC-MSCs CM. Patient received a single endovenous administration of phenylbutazone (Phenylarthrite^®^, 2.2 mg/kg, IV, Vetoquinol, France) at the end of the treatment. No other medical treatments (including nonsteroidal anti-inflammatory drugs, intra-articular corticosteroids, hyaluronan, glycosaminoglycans, hemoderivative treatments, and other MSCs preparations) were administered, except for those described in the treatment protocol, before and after allogenic eSM-MSCs + eUC-MSC CM treatment.

Patient was monitored for 48 h after treatment, and no complication was registered. Following the treatment, patient was assessed periodically to control swelling of the joint, lameness, and ultrasonographic changes (echogenicity, cross sectional area, and fiber alignment). Corrective asymmetrical shoeing with more support (wider branch) on the medial side was performed—“Denoix asymmetric shoe”. 

#### Intralesional eSM-MSCs + eUC-MSCs CM Administration

Patient was sedated with detomidine (Domosedan^®^, 0.02 mg/kg, IV, Orion Corporation, Finland), the right tarsus was trichotomized, and the skin was surgically disinfected with chlorohexidine and alcohol. The prepared therapeutic combination was aspired with a 18G needle to a 2 mL syringe and gently homogenized. Ultrasound-guided injection with a 20G needle was performed at the lesion site. The stablished protocol included a second eSM-MSCs + eUC-MSCs CM administration 15 days after the first treatment using the same protocol.

### 2.10. Post-Treatment Monitoring—Clinical Evaluations

Tissue regeneration was indirectly estimated through lameness evaluation, pain to pressure, limb inflammation, limb sensitivity, and ultrasound imaging. In each assessment, the ultrasonographic examination was performed in transverse and longitudinal scans, and three parameters were evaluated: lesion echogenicity, lesion longitudinal fiber alignment (FA), and cross-sectional area. The contralateral healthy limb was used as a control. Ultrasonographic evaluation was performed on assessment day, treatment day (day 1—T0), and days 15 (T1—second administration), 30 (T2), 45 (T3), 60 (T4), and 90 (T5) post treatment (Figure 2).

The rehabilitation program consisted of an exercise-controlled program including stall confinement and regular and increasing-time exercises, as presented in Table 2 [1,35,36,37,38]. Exercise was initiated early, on the second day after treatment, with low-level movements—hand walking. Early movements should include weight-bearing, strengthening, and flexibility activities, whereas stall rest alone should be used as infrequently as possible [36]. 

After eSM-MSCs + eUC-MSCs CM treatment, the patient underwent a rehabilitation program consisting of two days of box rest followed by 13 days of 10 min hand-walk. Bandage applied on the treatment day was removed 24 h after treatment. On day 15, the second treatment was performed, followed by the same 15-day rehabilitation program, until day 30. Between day 30 and day 45, the work consisted of 20 min hand-walking; between day 45 and day 60, the work was 30 min of hand-walking; between day 60 and day 75, the work consisted of 30 min of hand walking plus 5 min of trotting; and, finally, between day 75 and day 90, the patient underwent 30 min of hand-walking plus 10 min of trotting.

Veterinary assessment on day 90 (T5) determined if the horse could return to regular work based on clinical recovery (limb sensitivity and lameness evaluation) and ultrasonographic lesion improvement, evidenced by normal echogenicity, good fiber alignment, and normal cross-sectional area of the ligament when compared with contralateral limb.

## 3. Results

### 3.1. Clinical Evaluation

On the first assessment day, there was no pain at palpation and manipulation, and no swelling in the distal limb was observed. The horse was not lame at the walk or trot in a straight line but was a little reluctant to fully bear weight on the right hind leg when turned to the right (grade 2/5 according to AAEP lameness grading scale). Flexion test and pain to pressure were also evaluated, and no flexion response and pain to pressure were identified. 

### 3.2. Complementary Diagnostic Examinations

#### 3.2.1. Radiological Examination

A radiological examination was performed. The horse did not present significant articular abnormalities within the tarsocrural joint. There was tarsocrural joint distension, soft tissue distension, and slight evidence of tissue thickening at the injured long medial collateral ligament. Radiographs are presented in Figure 3. X-ray examination was not carried out again during the follow-up period.

#### 3.2.2. Ultrasound Examination

At the first veterinary assessment, ultrasonographic examination evidenced an increased amount of hypoechoic fluid containing areas of increased swirling (heterogeneous echogenicity) (Figure 4a), suggestive of an organized hematoma and/or fibrin within the joint—hemarthrosis. Also evidenced was a moderate fiber pattern disruption of LMCL (hypoechoic region) at medial malleolus insertion as well as an increased cross-sectional area (Figure 4b). Cartilage surface was normal. 

### 3.3. MSCs Isolation and Characterization

Equine SM-MSCs were fully characterized, with the goal of confirming the presence of the minimum classification criteria necessary to classify their identity as MSCs, as previously described [4,5,6]. Briefly, the plastic-adherence of the cells when in culture as well as their fibroblast-like morphology were confirmed. The tri-differentiation capacity was confirmed by exposing eSM-MSCs to specific media to induce adipogenic, chondrogenic, and osteogenic differentiation. In addition to the morphological changes, the differentiations were confirmed qualitatively through staining with Oil Red O, Alcian Blue, and Alizarin Red, respectively. These cells were previously characterized by immunocytochemically, and positive staining was identified for markers indicating stemness (Octamer-binding transcription factor 4 (OCT4) and Homeobox protein NANOG (NANOG)), mesenchymal/stromal histogenesis (vimentin), and synovial histogenesis (lysozyme). Karyotype evaluation revealed chromosomal structural normality, normal number of chromosomes (64, XY), and absence of neoplastic alterations. The bacteriological control confirmed the absence of bacterial growth in both aerobiosis and anaerobiosis, and also the absence of fungal contamination after five days of incubation.

Equine UC-MSCs have been successfully isolated from UC tissue (Figure 5a). Cells in culture presented clear plastic-adhesion and fibroblast-like morphology (Figure 5b).

Equine UC-MSC trilineage differentiation was confirmed: adipogenic differentiation was observed by the presence of large red-stained lipid vacuoles in the cytoplasm due to exposure of Oil Red O staining; chondrogenic differentiation was observed by the presence of proteoglycans-marked deposition in the extracellular matrix, which was stained blue, confirming the presence of chondrogenic aggregates; and osteogenic differentiation was demonstrated by the presence of extracellular phosphate deposits brown-stained by Von Kossa solution (Figure 6).

The eUC-MSCs showed strong expression in vimentin, confirming their mesenchymal/stromal and non-epithelial origin and no expression of pan-cytokeratin (AE1 AE3); and a platelet endothelial cell adhesion molecule (CD31) which confirms the non- epithelial and non-vascular histogenesis of the cells, respectively. Altogether, these results confirm the expected mesenchymal/stromal origin of the cells (Figure 7).

### 3.4. Secretome: Conditioned Medium Analysis

The results indicated that IL-6 and IL-8 were produced in high levels ranging from IL-6: 29.66 ± 2.64 pg/mL to IL-8: 14.05 ± 0.96 pg/mL. The average concentration quantified for each interleukin is depicted in Appendix A. 

### 3.5. Treatment Results

The patient did not present any adverse event that required treatment cessation, unplanned procedures, or additional treatments. The two intra-ligamentous administrations and follow-up procedures did not display adverse reactions (inflammation, infection, deterioration of the lesion, increased lameness), neither at treatment time (T0 and T1) nor during the following weeks.

On day 30 (T2), there was no evidence of pain and lameness (grade 0/5). Ultrasonographic evaluation evidenced increased echogenicity of the lesion as well as a reduction in the cross-sectional area with good fiber alignment. The tarsocrural joint swelling and oedema were reduced. Nevertheless, compared with the contralateral limb, the right tarsocrural joint diameter was still larger than the left. 

Over the course of the follow-up ultrasonographic examinations, an increasing echogenicity of the lesion was evidenced, as well as a reduction in the cross-sectional area, good fiber alignment, and a reduction of the abnormal synovial fluid. On day 60 (T4), two months after the first treatment, there was a complete recuperation of the ligament structure—lesion completely fulfilled, good echogenicity, good fiber alignment, and normal cross-sectional area—compatible with adequate tissue regeneration. No pain and no lameness were present, and there were also no signs of cartilage remodeling. Despite this achievement, a physical rehabilitation program proceeded until day 90 (T5).

To sum up, the assessment of the patient’s clinical recovery was performed by the presence of no pain in the tarsus and no lameness achieved by day 30. Lesion ultrasonographic improvement and indicators of regeneration were evidenced by a progressive increase in echogenicity and fiber alignment, and decrease in ligament cross-sectional area and synovial fluid accumulation within the joint space, observed through ultrasound examination during the follow-up on days 1 (T0), 15 (T1), 30 (T2), 45 (T3), and 60 (T4) (Figure 8).

On day 90 (T5), the patient returned to regular work with no lesion relapse reported up to 18 months after injury. Additional information reports that patient is already participating in competitions at a higher level than before the injury.

## 4. Discussion 

This research discloses the application of a treatment strategy that focuses on the follow-up and healing features of an equine injured ligament, as well as on the patient rehabilitation and return to competition. Herein, we also highlight the effect of combining eSM-MSCs and eUC-MSCs populations in the treatment, using either the cells themselves or the cell-derived secretome.

In the current study, the use of the therapeutic combination of eSM-MSC + eUC-MSC CM in the treatment of LMCL desmitis was considered successful as clinical and ligament ultrasonographic parameters returned to normal.

On day 30 (T2), no pain and lameness (Score 0/0, AAEP Lameness Score) were registered. On day 45 (T3), there was an impressive ultrasonographic recovery, and on day 60 (T4), a complete ultrasonographic ligament recovery with the achievement of good fiber alignment, normal cross-sectional area, and good echogenicity was observed. It must be highlighted that, usually, with conventional treatments, only scar tissue repair occurs, and regeneration is never accomplished. In addition, full ultrasonographic recovery time was achieved in 50% of that described in the literature, being considered successful [5,6].

Clinically, on day 90, the patient was able to resume full work. This period of time also corresponds to half of that reported in literature when different therapeutic protocols are used in the same type of lesion. On the other hand, in the present case, there was a significative reduction in the rest period. In 2 days, this horse started the rehabilitation program versus 30–180 days of rest, and it returned to full work after 90 days versus 180 days presented in other studies, which accounts for almost 50% of the reduction in the total recovery time [6,9,18]. A study referred to the treatment of this type of lesion with platelet-rich plasma (PRPs), achieving a return to the same level of work in 180 days in 81% of the horses versus 90 days with this combination product [18]. The recovery times obtained in the current case remain a great attainment compared with those described by others concerning equine clinical trials of desmitis of collateral ligament of tarsus and equine tendonitis [5,6,7,8,9]. 

Remarkably, after the rehabilitation program was completed, the patient returned to the same physical work conditions and even exceeded the previous performance levels, exceeding the usual jumping level from 1.0 m to 1.30 m. 

It is important to highlight that no side effects were registered during the treatment and rehabilitation periods, and 18 months after injury, no lesion relapse was noteworthy. Nevertheless, a slight distension of the right tarsus grossly remains currently perceptible when compared with the contralateral limb, and, ultrasonographically, the affected joint also presents sparse oedema. 

To the best of our knowledge, there are no references to LCML healing with recovering of the normal ligament ultrasonographic features and absence of clinical signs in such a short period of time. Prognosis for medial tarsal collateral ligament desmitis appears good for survival but fair for return to previous levels of performance and requires prolonged periods of rest and a controlled exercise program [39]. 

Importantly, besides the effective result, there are other advantages concerning this treatment protocol: its allogenic origin and easy application and access. The allogeneic source is a key factor in this therapeutic product with no observed adverse or rejection reactions supporting its potential as an alternative to autologous therapies, promoting its ready-to-use application. Furthermore, the existence of a cell and secretome bank that enables the production and validation of both MSCs and secretome, and offers well-characterized orthobiological products with recognized benefits in musculoskeletal tissue regeneration which might be stored and readily available for administration, becomes clinically very appealing and advantageous. This allows for early medical intervention in acute cases based on prompt and easy procedures such as an injection, contributing to a better functional outcome and a rapid and sustainable return to the sportive career. 

As previously reported, SM-MSCs and UC-MSCs present a high tenogenic, anti-inflammatory, multipotency, and low-immunogenicity abilities, being, therefore, very appealing for tendon and ligament regeneration [29,40,41,42]. Equine SM-MSCs secrete high levels of human growth-regulated oncogene/keratinocyte chemoattractant (KC/GRO), monocyte chemoattractant protein-1 (MCP-1), IL-6, basic fibroblast growth factor (FGF-2), granulocyte colony-stimulating factor (G-CSF), granulocyte macrophage colony-stimulating factor (GM-CSF), and IL-8 [28]. This profile supports their reported benefits in fibroblast-intense activity and lesion reperfusion, proliferation of tenogenic stem cells, enhancing cell proliferation and collagen production [43]. Other factors such as G-CSF and GM-CSF also depict potential as skeletal muscle repair mediators, including those with pro-inflammatory functions [44,45]. Pro-inflammatory factors such as those found in these cells secretomes (GM-CSF, G-CSF, Il-6, IL-8 and IL-17) are frequently regarded as deleterious. However, they are involved in damage signaling and subsequent activation of resident tendon cells for effective healing, stimulating tendon cell proliferation [46,47].

The current study also demonstrated the presence of high levels of IL-6 and IL-8 in eUC-MSC CM. However, the exact meaning of this finding remains to be clarified.

Interleukin-6 bares pro-inflammatory and angiogenic functions, capable of increasing the expression of other growth factors (GF). Immunosuppressive properties are also associated with IL-6, which may be prime motors for the success of allogenic MSC implantation [48,49]. This pro-inflammatory nature is associated with the induction of acute-phase proteins, inducing a potent regeneration of various tissues and supporting their potential as a therapeutic approach for regenerative medicine [50,51]. Previous studies have, likewise, demonstrated that IL-6 is a potent anti-inflammatory cytokine significantly up-regulated in injured human tendons [52]. This cytokine has been demonstrated to have an important role in regulating tendon-derived stem cell (TDSC) activity and differentiation, however inhibiting their tenogenic differentiation in vitro [53], while in an in vivo model (IL-6 −/− mice), it has been demonstrated to be involved in the complex mechanisms that contribute to mechanical and organizational properties of injured tendons [54]. Another in vivo study demonstrated that a human Achilles tendon presented high levels of various growth factors after exercise. Among these, IL-6 was largely present, suggesting a responsibility in transforming collagen under biomechanical stimulation. After an experimental infusion of IL-6 in the peritendinous tissue, followed by exercise, collagen synthesis stimulation was observed, corroborating the hypothesis that IL-6 is an important growth factor for connective tissue in healthy human tendons [55].

Interleukin-8 is also a recognized pro-inflammatory mediator and a potent angiogenic factor associated with the increase in vascular endothelial growth factor (VEGF) concentration. Interleukin-8 was directly related to VEGF stimulation, helping in the revascularization and ligamentization of a grafted tendon [56]. Interleukin-8 has a similar effect to IL-6 but has a longer half-life [57]. 

A study on human Achilles tendon showed that IL-6, IL-8, and IL-10 were upregulated in a tendon healing phase with the absence of inflammation, indicating that these cytokines may be associated with anti-inflammatory and regenerative activity in the tendon healing process [52].

Herein, the high in vitro production of IL-6 and IL-8 by the MSCs populations under study suggests a putative involvement and contribution of these bioactive molecules in diverse biological functions related to immunomodulative and regenerative processes, magnifying the potential benefits of this therapeutic combination [58].

Hypothetically, eSM-MSCs and eUC-MSCs secretome factors are able to promote tendon/ligament healing by stimulating fibroblastic and angiogenic proliferation, reactivating growth programs, reducing inflammation, and stimulating cell proliferation, collagen production, and tenogenic differentiation, accomplishing not only lesion repair with regenerated tissue but also strengthening of the entire ligament, reducing the risk of lesion recurrence [39]. Nevertheless, further research is needed to more accurately understand the real influencing role of IL-6 and IL-8 in vivo. Having this potential in mind, one might believe that the combined action of eSM-MSCs and UC-MSC CM was a therapeutical advantage in this study case. 

As typical of other studies, the current also has limitations, and the most important is related to the fact that only one animal case is presented herein, impairing conclusions about the real benefits of the therapeutic product tested. Also, this work may contain some bias as it refers to a specific patient treated with a certain batch of cells and a CM preparation that might present some variations in its composition in other cases. In this way, further investigations involving a larger number of patients in more controlled and standardized clinical conditions should be carried out to prove the effectiveness of this treatment. 

From an ethical perspective, it is also significant to state that, in the particular context of orthopedic research, many studies can be conducted in naturally occurring disease (without premeditated disease induction) and that the horse often poses as both the model and final beneficiary of the developed therapies, alleviating the ethical burden of such studies.

## 5. Conclusions

The outcome of the use of the therapeutic combination of eSM-MSCs and eUC-MSCs CM in the treatment of the presented clinical case of desmitis of LCML was considered successful. The clinical re-evaluations showed a fast, efficient, and safe clinical recovery with very positive outcomes: no lameness, and ultrasonographic images were compatible with regeneration (lesion fulfillment, parallel fibers, and normal cross-sectional area). The patient returned to a sportive career, reaching higher jump performance levels. After 18 months, there were no evidences of lesion relapse.

Clinical injuries of LMCL are very difficult to treat, in part due to their frequent misdiagnosis as well as their long-term recovery, meaning outcomes have frequently poor prognosis in terms of competition return. The fact of having a complete clinical recovery and ultrasound images compatible with LMCL regeneration in 60 days remains a very encouraging and promising accomplishment. Nevertheless, whether this result is due to the eSM-MSCs or to the eUC-MSCs CM or their patented combination cannot be fully determined since this study reports results of only one patient. Therefore, further research into the specific mechanisms of therapeutic action and clinical trials are required to further validate the approach and confirm the real benefits of this combination. 

## Figures and Tables

**Figure 1 animals-14-00370-f001:**
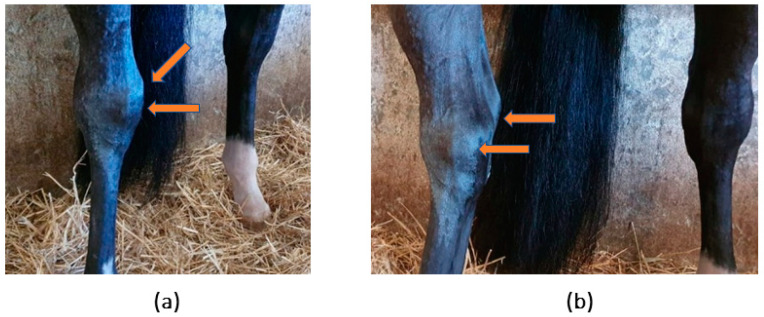
Horse’s clinical inspection. Evidence of increased volume of the right tarsocrural joint. (**a**) Frontal view and (**b**) medial view.

**Figure 2 animals-14-00370-f002:**
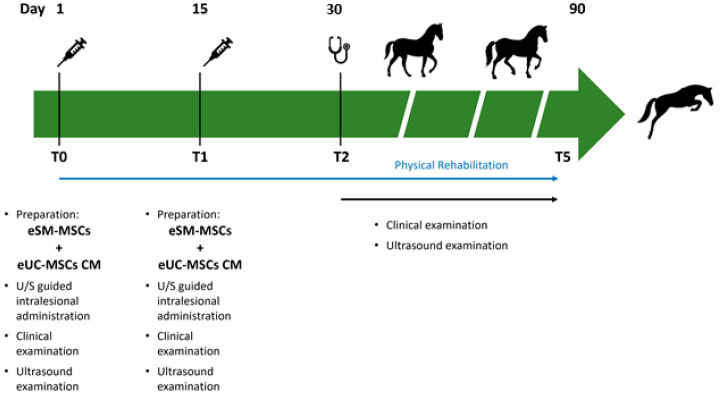
Timeline of eSM-MSCs treatment protocol and rehabilitation program. T0 is the day of the first treatment with the administration of eSM-MSCs + eUC-MSCs CM combination. Besides the intralesional application of the therapeutic combination, clinical and ultrasound examinations were also performed. T1 refers to the second application of the composition 15 days after T0, when the same procedure was repeated. On the other assessment days—T2, T3, T4, and T5—clinical and ultrasound examinations were performed. At the same time, a physical rehabilitation plan was carried out.

**Figure 3 animals-14-00370-f003:**
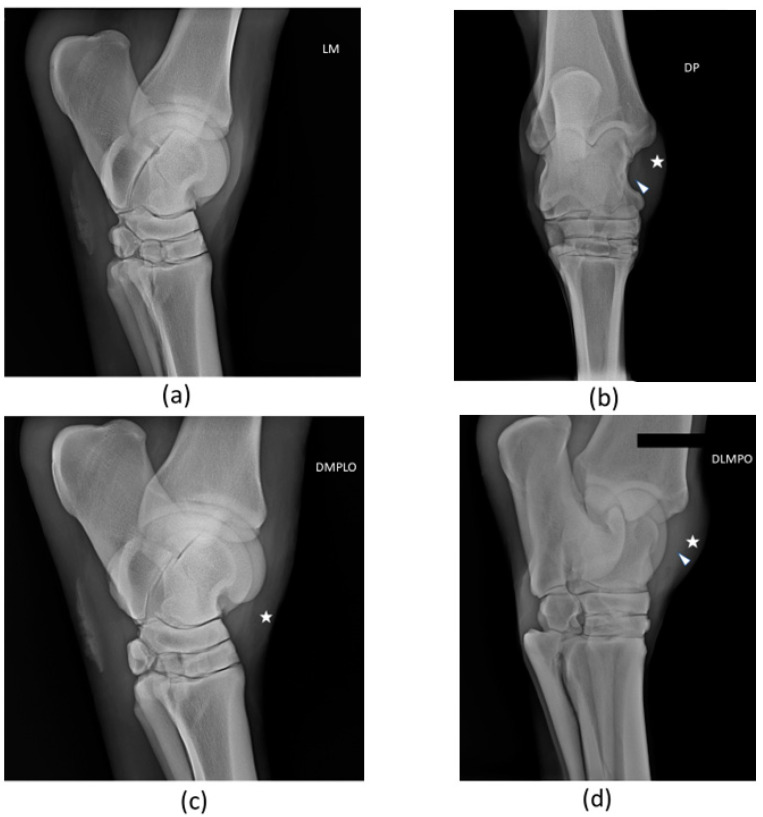
Radiographic views of patient’s right tarsus: (**a**) Lateromedial (LM), (**b**) Dorsoplantar (DP), (**c**) Oblique dorsomedial-plantarolateral (DMPLO), (**d**) Oblique dorsolateral-plantaromedial (DLPMO). The white head of the arrow (△) points to increased radiopacity of the long medial collateral ligament, and the star (*) signals soft tissue swelling and joint distension. There are no significant radiological alterations of articular surfaces.

**Figure 4 animals-14-00370-f004:**
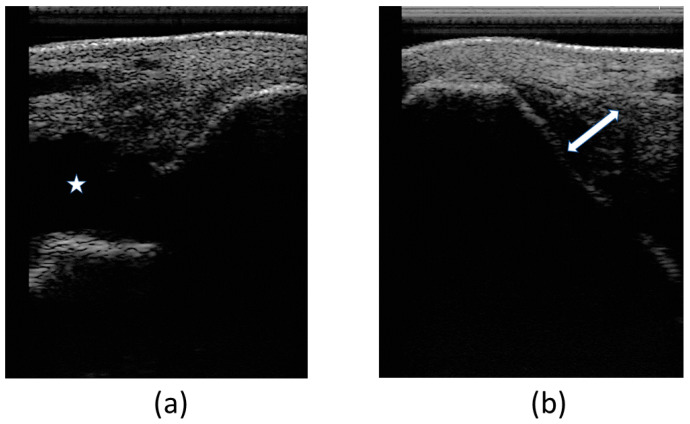
Images of the first ultrasonographic examination. Desmitis of LMCL insertion at the medial malleolus: (**a**) increased amount of hypoechoic fluid within the joint, signaled with the star (*); (**b**) disruption of the fibers at the insertion, signaled with the double arrow (↔).

**Figure 5 animals-14-00370-f005:**
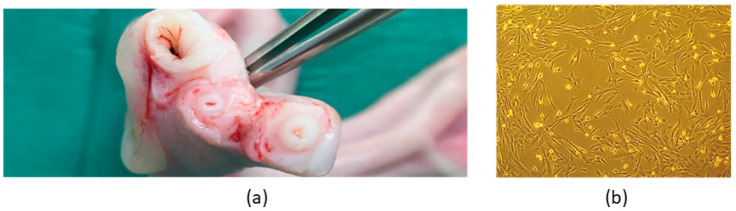
Isolation of MSC from equine umbilical cord tissue. (**a**) Umbilical cord tissue. (**b**) Isolated population of eUC-MSCs at P3—plastic adhesion, monolayer, and fibroblast-like shape of eUC-MSCs may be observed.

**Figure 6 animals-14-00370-f006:**
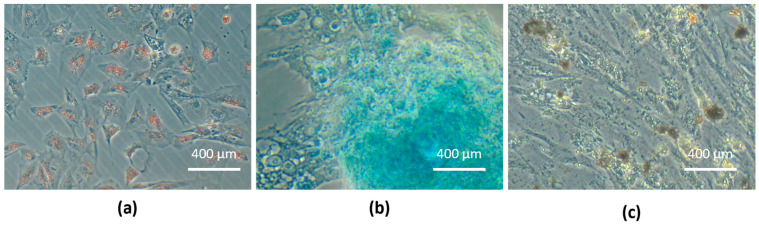
Tri-lineage differentiation: (**a**) eUC-MSCs adipogenic differentiation, cytoplasmatic lipid vacuoles stained in red (Oil Red O stain); (**b**) eUC-MSCs chondrogenic differentiation, proteoglycans in extracellular matrix stained in blue (Alcian Blue staining); (**c**) eUC-MSCs osteogenic differentiation, extracellular phosphate deposits stained in brown (Von Kossa staining).

**Figure 7 animals-14-00370-f007:**
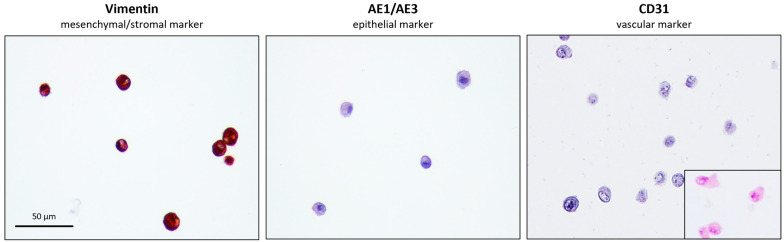
Immunocytochemistry of eUC-MSC. Positive staining for mesenchymal/stromal marker vimentin (**left)** and negative for the epithelial marker AE1/AE3 (**middle**) and vascular marker CD31 (**right**), H-E staining (lower right insert).

**Figure 8 animals-14-00370-f008:**
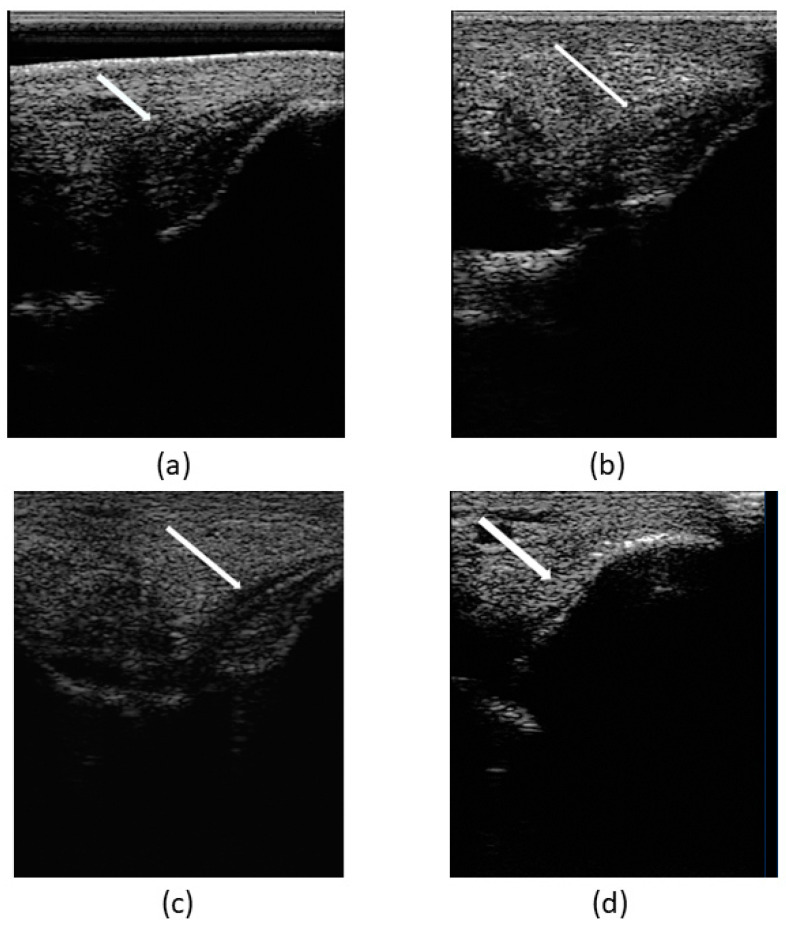
Images of ultrasonographic follow-up. (**a**) Day 1 (T0), (**b**) day 15 (T1), (**c**) day 30 (T2), and (**d**) day 60 (T3). Indicators of ligamentous regeneration: increased echogenicity and fiber alignment, as well as decrease in cross-sectional area and synovial fluid accumulation within the joint space.

**Table 1 animals-14-00370-t001:** Score systems used to assess lameness and response to flexion test [34].

Parameter	Score	Clinical Implication
AAEP Grading	0	No Lameness
	1	Lameness not consistent
	2	Lameness consistent under certain circumstances
	3	Lameness consistently observable on a straight line
	4	Obvious lameness at walk: marked nodding or shortened stride
	5	Minimal weight-bearing lameness in motion or at rest
Flexion Test	0	No flexion response
	1	Mild flexion response
	2	Moderate flexion response
	3	Severe flexion response

**Table 2 animals-14-00370-t002:** Physical rehabilitation program.

Week	Exercise
0–2	2 days: stall confinement Handwalk: 10 min Day 15: new treatment
3–4	2 days: stall confinement Handwalk: 10 min VET-CHECK + U/S
5	Handwalk: 15 min
6	Handwalk: 20 min VET-CHECK + U/S
7	Handwalk: 25 min
8	Handwalk: 30 min VET-CHECK + U/S
9–10	Handwalk: 30 min + 5 min trot
11–12	Handwalk: 30 min + 10 min trot VET-CHECK + U/S

## Data Availability

Data are contained within the article and Appendix A.

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
