# Peer review of "Treatment of Equine Tarsus Long Medial Collateral Ligament Desmitis with Allogenic Synovial Membrane Mesenchymal Stem/Stromal Cells Enhanced by Umbilical Cord Mesenchymal Stem/Stromal Cell-Derived Conditioned Medium: Proof of Concept"

_animals, 2024, doi:10.3390/ani14030370_

Round 1
Reviewer 1 Report (Previous Reviewer 1)
Comments and Suggestions for Authors
I carefully read the revised version of the manuscript in object and the Authors’ comments.
Most of my suggestions have been completely or partially addressed and, in my opinion, the impact of the manuscript is increased.
First of all, turning type of the manuscript to “original article” and presenting the clinical case as a “proof-of-concept” was correct.
Nonetheless, I still find the Introduction over-written with too much detailed description of the field of knowledge. Once again, as the paper is directed to experts in the field and not educational, much information about anatomy and physiology as well as other basic knowledge (including educational figures as Fig 1 and 2) could be significantly shortened or even omitted.
Similarly, the extended description of the state-of-the-art in the Discussion section is still over-written, to me. Discussion should be focused on data presented in comparison with the latest publication in the field demanding a brief description of knowledge in the field to the Introduction section.
In addition, a short discussion on the limits of the present work should be included. In fact as a proof-of-concept work it could bring important bias as; application of different batches of cell and CM preparation, different cohort of treated patients and so on.
In my opinion, this could strongly improve the readability of the manuscript and the intelligibility of the data presented.
Author Response
Responses to the Reviewers´ Comments and Suggestions
Dear Members of Editorial Office,
Animals
Title: Treatment of Equine Tarsus Long Medial Collateral Desmitis with allogenic synovial membrane mesenchymal stem cells enhanced by umbilical cord mesenchymal stem cell-derived conditioned medium: case report
Authors: Inês Leal Reis, Bruna Lopes, Patrícia Sousa, Ana Catarina Sousa, Mariana Branquinho, Ana Rita Caseiro, Alexandra Rêma, Inês Briote, Carla Mendonça, Jorge Miguel Santos, Luís Atayde, Rui Alvites, Ana Colette Maurício
We are sending the revised form of the manuscript entitled “Treatment of Equine Tarsus Long Medial Collateral Desmitis with allogenic synovial membrane mesenchymal stem cells enhanced by umbilical cord mesenchymal stem cell-derived conditioned medium: case report”
The authors addressed, point by point, all the concerns of the reviewers and associate editor. The changes are shown in the marked version of the manuscript (using track changes mode in MS Word highlighted in red).
We hope that this revised version of the manuscript fulfils all the requirements in order to be published in the Animals journal.
On behalf of all authors,
Ana Colette Maurício
Reviewer 1
First of all, turning type of the manuscript to “original article” and presenting the clinical case as a “proof-of-concept” was correct.
The authors thank you for agreeing with the changes introduced.
Nonetheless, I still find the Introduction over-written with too much detailed description of the field of knowledge. Once again, as the paper is directed to experts in the field and not educational, much information about anatomy and physiology as well as other basic knowledge (including educational figures as Fig 1 and 2) could be significantly shortened or even omitted.
The introduction was revised. Images, as well as the anatomy and physiology section, were excluded. The main text was rewritten.
Similarly, the extended description of the state-of-the-art in the Discussion section is still over-written, to me. Discussion should be focused on data presented in comparison with the latest publication in the field demanding a brief description of knowledge in the field to the Introduction section.
The section was revised.
In addition, a short discussion on the limits of the present work should be included. In fact as a proof-of-concept work it could bring important bias as; application of different batches of cell and CM preparation, different cohort of treated patients and so on.
The section revised and a small paragraph was included to overcome this weakness. The text segment is highlighted in yellow.
However, it must be considered that this "proof of concept" refers only to a clinical case and that it may present some bias such as this work refers to a specific patient, treated with a certain batch of cells and a CM preparation that might present some variations in its composition in another cases. In this way, we are sure that more stud-ies should be done to prove the effectiveness of this treatment.
In my opinion, this could strongly improve the readability of the manuscript and the intelligibility of the data presented.
Thank you for your input.

Reviewer 2 Report (Previous Reviewer 3)
Comments and Suggestions for Authors
The work is interesting as it is focused on the clinical outcome of the treatment of long medial collateral ligament desmitis in the tarsal joint. The manuscript is the case report of the first use of allogenic synovial membrane MSCs enhanced by umbilical cord MSCs in the specific treatment protocol. As an attempt to describe a new direction in treatment, the manuscript deserves recognition, but the presentation of the results leaves much to be desired. Please see detailed comments that are intended to significantly improve the quality of the presentation of your results.
The manuscript requires revision following the Instructions for Authors published on https://www.mdpi.com/journal/animals/instructions
Please see that case reports: must provide an in-depth, rather than superficial, review of a particular case. The purpose of this presentation should be to suggest a novel way of interpreting existing knowledge on the topic. Articles are usually identified as case studies rather than scientific papers if they contain material that is deemed to be open to interpretation, rather than empirically proven. Case studies, like Commentaries, must contain reasoned arguments, not just arguments expressing the emotions of the author. Case studies must be concise, well-argued, and erudite, as well as be written to a very high standard of English. All submissions will be submitted for peer review. Case reports should have a main text of around 2500 words.
Simple Summary sections should appear in all manuscript types.
Three to ten pertinent keywords need to be added after the abstract.
References should be described as follows, depending on the type of work:
Journal Articles:
1. Author 1, A.B.; Author 2, C.D. Title of the article. Abbreviated Journal Name Year, Volume, page range.
Books and Book Chapters:
2. Author 1, A.; Author 2, B. Book Title, 3rd ed.; Publisher: Publisher Location, Country, Year; pp. 154–196.
3. Author 1, A.; Author 2, B. Title of the chapter. In Book Title, 2nd ed.; Editor 1, A., Editor 2, B., Eds.; Publisher: Publisher Location, Country, Year; Volume 3, pp. 154–196.
Since this is the second time you are submitting the same article to the same journal, all author guidelines should be followed.
General comments
Your manuscript is definitely too long and does not fit the set word limit for Case reports. One possible solution to this problem is to transfer some of the information to supplementary materials or to divide this very extensive study into two good, concise works with different goals. One article may actually be a good clinical case (if you follow the guidelines for Case reports) and the other a very good methodological study (if it focuses on cell culture and MSCs preparation protocols). Possibly the latter (MSCs preparation protocols can be entirely transferred to supplementary materials - both from the M&M and results sections).
The simple summary is missing.
Too many keywords are used.
The introduction section is too long and does not focus on introducing the topic. The introduction section should be shortened and rewritten according to the detailed comments.
The aim of the study requires rewriting.
The M&M and results sections require changes described in detailed comments.
The discussion section should be strongly restructured and shortened to focus just on the aim of the study. In its current form, no apparent logical order is used in comparing/contrasting the results obtained in the study with previous literature. The authors should attempt structuring the discussion following "the common standard format" which usually consists of the following points:
a. One sentence summary that highlights the most relevant results.
b. A thorough discussion of each result obtained concerning the corresponding study objective: was the tested hypothesis confirmed or not? Why? What previous evidence supports the specific result or not? It is critical to compare/contrast the result obtained with previous literature in the equine species first, then in veterinary medicine, and finally in human medicine (if not enough data are available for comparison in veterinary medicine
c. Statement of study limitations
d. Future directions
The conclusions section should be thoroughly revised.
The References section is prepared incorrectly - please carefully check the Instructions for Authors and the method of citing references. This entire section needs improvement. Also, references contained in the manuscript body require improvement and unification - a numerical system is adopted, and references addressed by italics require correction.
Detailed comments
L 1 Change "Original Research Manuscript" to "Case report"
L 48-62 and L 83-92 connect and shorten these paragraphs to 6-8 sentences about tarsal ligament desmitis. Remove bold font.
L 61 remove Figure 1
L 63-72 An introduction to the anatomy of the ankle ligaments is unnecessary - remove the entire paragraph
L 73-76 The range of motion of the tarsocrural joint is not the aim of the study - remove the entire paragraph
L 93-108 This is not the aim of your study - remove the entire paragraph
L 109- 122 The recently available treatment protocols of tarsal ligament desmitis should be presented briefly
L 123-172 The whole part on regenerative medicine should be shortened to no more than half a page and briefly focused on the strategy of using synovial membrane MSCs enhanced by umbilical cord MSCs
L 174 remove Figure 2
L 178 expend abbreviation LMCL. (All abbreviations, including cytokine names (L 185), should be expanded when using them for the first time).
L 177-179 I agree with the leading aim of the study. However, the second (L 180-186) can not be supported using the single case report. I recommend that you reword the entire manuscript to focus on the first aim. If this work is to remain a case report, please remove the second aim of the study.
L 221 add the distance between the X-ray tube and the cassette.
L 238 If this work is to remain a case report, please move sections 2.5, 2.6, 2.7, 2.8, and 2.9 to supplementary materials or a new article. Continue the report case description with 2.10 and 2.11 subsections (connect seb-subsection 2.10.1 to subsection 2.10)
L 323-327 I still don't understand the statistical analysis you presented - please provide the raw data you compare (taking into account the number of repetitions and examining the distribution of features).
L 414 The results section should be written to collate detailed clinical and radiological examination results of the horse before treatment and at T3, T4, and T5 so that the progression of changes in each clinical (including AAEP grading) and radiological signs can be tracked.
L 437 If this work is to remain a case report, please move sections 3.2 and 3.3 to supplementary materials or a new article.
L 505 The presentation and preparation of the data series for this comparison is completely incomprehensible to me. Please present the raw data, describe how you obtained it what you compare it with, and for what purpose.
L 542 After implementing all the suggested changes, the results obtained might significantly differ from those currently reported in the manuscript. Thus, no specific comments are provided below, and a more thorough review of the discussion section will be provided once the manuscript is considered overall scientifically sound.
L 705 Conclusions should be thoroughly revised and should be closely related to the purpose of the work. Since it was recommended to limit the purpose of the study only to the description of a clinical case, the entire conclusions section should be rewritten. Unfortunately, there is no room for speculation that is not supported by the research results described in the manuscript.
Author Response
Responses to the Reviewers´ Comments and Suggestions
Dear Members of Editorial Office,
Animals
Title: Treatment of Equine Tarsus Long Medial Collateral Desmitis with allogenic synovial membrane mesenchymal stem cells enhanced by umbilical cord mesenchymal stem cell-derived conditioned medium: case report
Authors: Inês Leal Reis, Bruna Lopes, Patrícia Sousa, Ana Catarina Sousa, Mariana Branquinho, Ana Rita Caseiro, Alexandra Rêma, Inês Briote, Carla Mendonça, Jorge Miguel Santos, Luís Atayde, Rui Alvites, Ana Colette Maurício
We are sending the revised form of the manuscript entitled “Treatment of Equine Tarsus Long Medial Collateral Desmitis with allogenic synovial membrane mesenchymal stem cells enhanced by umbilical cord mesenchymal stem cell-derived conditioned medium: case report”
The authors addressed, point by point, all the concerns of the reviewers and associate editor. The changes are shown in the marked version of the manuscript (using track changes mode in MS Word highlighted in red).
We hope that this revised version of the manuscript fulfils all the requirements in order to be published in the Animals journal.
On behalf of all authors,
Ana Colette Maurício
Reviewer 2
L 1 Change "Original Research Manuscript" to "Case report"
the authored decided to adapt the article as a “Proof of concept” manuscript because, considering the exploratory nature of the work carried out.
L 48-62 and L 83-92 connect and shorten these paragraphs to 6-8 sentences about tarsal ligament desmitis. Remove bold font.
The proposed changes have been introduced.
L 61 remove Figure 1
The proposed changes have been introduced.
L 63-72 An introduction to the anatomy of the ankle ligaments is unnecessary - remove the entire paragraph
The proposed changes have been introduced.
L 73-76 The range of motion of the tarsocrural joint is not the aim of the study - remove the entire paragraph
The proposed changes have been introduced.
L 93-108 This is not the aim of your study - remove the entire paragraph
The proposed changes have been introduced.
L 109- 122 The recently available treatment protocols of tarsal ligament desmitis should be presented briefly.
The proposed changes have been introduced.
L 123-172 The whole part on regenerative medicine should be shortened to no more than half a page and briefly focused on the strategy of using synovial membrane MSCs enhanced by umbilical cord MSCs
The proposed changes have been introduced.
L 174 remove Figure 2
The proposed changes have been introduced.
L 178 expend abbreviation LMCL. (All abbreviations, including cytokine names (L 185), should be expanded when using them for the first time).
The proposed changes have been introduced.
L 177-179 I agree with the leading aim of the study. However, the second (L 180-186) can not be supported using the single case report. I recommend that you reword the entire manuscript to focus on the first aim. If this work is to remain a case report, please remove the second aim of the study.
As explained in the answer to the first issue raised by the reviewer, the authors understood the fragility of the article and chose to classify it and rewrite it as a proof of concept.
L 221 add the distance between the X-ray tube and the cassette.
The x-ray tube establishes the distance itself with a laser beam so it did not seem to be an important factor for the authors. The laser beam has approximately 1 meter.
L 238 If this work is to remain a case report, please move sections 2.5, 2.6, 2.7, 2.8, and 2.9 to supplementary materials or a new article. Continue the report case description with 2.10 and 2.11 subsections (connect seb-subsection 2.10.1 to subsection 2.10)
This is a “proof of concept” article.
L 323-327 I still don't understand the statistical analysis you presented - please provide the raw data you compare (taking into account the number of repetitions and examining the distribution of features).
As described in the article, the statistical analysis was only carried out to evaluate the concentration of biomarkers present in the secretome of the two cells in comparison, with the samples being evaluated in triplicate. Interleukin concentrations were studied using the ANOVA test, and statistical differences were considered when and significance defined for P<0.05.
L 414 The results section should be written to collate detailed clinical and radiological examination results of the horse before treatment and at T3, T4, and T5 so that the progression of changes in each clinical (including AAEP grading) and radiological signs can be tracked.
The proposed changes have been introduced. Radiological examination was not the target of the revaluations as it is described at methods. Ultrasonographic examinations were the focus as well as lameness.
L 437 If this work is to remain a case report, please move sections 3.2 and 3.3 to supplementary materials or a new article.
Please, read the answers above.
L 505 The presentation and preparation of the data series for this comparison is completely incomprehensible to me. Please present the raw data, describe how you obtained it what you compare it with, and for what purpose.
The proposed changes have been introduced.
L 542 After implementing all the suggested changes, the results obtained might significantly differ from those currently reported in the manuscript. Thus, no specific comments are provided below, and a more thorough review of the discussion section will be provided once the manuscript is considered overall scientifically sound.
The proposed changes have been introduced.
L 705 Conclusions should be thoroughly revised and should be closely related to the purpose of the work. Since it was recommended to limit the purpose of the study only to the description of a clinical case, the entire conclusions section should be rewritten. Unfortunately, there is no room for speculation that is not supported by the research results described in the manuscript.
The proposed changes have been introduced.

Round 2
Reviewer 2 Report (Previous Reviewer 3)
Comments and Suggestions for Authors
Dear authors,
I appreciate your work and the concept and general idea of the publication. However, it does not meet the criteria for an Original Research Manuscript at this time.
I will also refer, point by point, to the answers of the Authors from the first round of revision. The authors declare that the changes are shown in the marked version of the manuscript (using track changes mode in MS Word highlighted in red), however, I didn't find any changes marked this way. Maybe this is the reason for my confusion. Please make sure the correct file is attached to the submission system.
If not, please submit the correct file. If yes, please see the comments listed below.
Previous comment "The manuscript requires revision following the Instructions for Authors published on https://www.mdpi.com/journal/animals/instructions (...)"
This comment has not been addressed. The manuscript style has not been corrected.
Previous comment "Simple Summary sections should appear in all manuscript types".
Previous comment "The simple summary is missing".
This comment has not been addressed. The simple summary has not been added.
Previous comment "References should be described as follows, depending on the type of work (...)"
"The References section is prepared incorrectly - please carefully check the Instructions for Authors and the method of citing references. This entire section needs improvement. Also, references contained in the manuscript body require improvement and unification - a numerical system is adopted, and references addressed by italics require correction."
This comment has not been addressed. The references has not been corrected.
Previous comment "Your manuscript is definitely too long and does not fit the set word limit for Case reports. One possible solution to this problem is to transfer some of the information to supplementary materials or to divide this very extensive study into two good, concise works with different goals. One article may actually be a good clinical case (if you follow the guidelines for Case reports) and the other a very good methodological study (if it focuses on cell culture and MSCs preparation protocols). Possibly the latter (MSCs preparation protocols can be entirely transferred to supplementary materials - both from the M&M and results sections). "
This comment has not been addressed.
The authors just stated that "the authored decided to adapt the article as a “Proof of concept” manuscript because, considering the exploratory nature of the work carried out". However, this manuscript does not meet the criteria for an original research manuscript.
Previous comment "The introduction section is too long and does not focus on introducing the topic. The introduction section should be shortened and rewritten according to the detailed comments."
This comment has been partially addressed.
Previous comment "The aim of the study requires rewriting."
This comment has not been addressed.
Previous comment "The M&M and results sections require changes described in detailed comments."
This comment was only partially addressed in the responses to detailed comments.
Previous comment "The discussion section should be strongly restructured and shortened to focus just on the aim of the study. In its current form, no apparent logical order is used in comparing/contrasting the results obtained in the study with previous literature. The authors should attempt structuring the discussion following "the common standard format" which usually consists of the following points:
a. One sentence summary that highlights the most relevant results.
b. A thorough discussion of each result obtained concerning the corresponding study objective: was the tested hypothesis confirmed or not? Why? What previous evidence supports the specific result or not? It is critical to compare/contrast the result obtained with previous literature in the equine species first, then in veterinary medicine, and finally in human medicine (if not enough data are available for comparison in veterinary medicine
c. Statement of study limitations
d. Future directions
The conclusions section should be thoroughly revised."
Previous comment "L 1 Change "Original Research Manuscript" to "Case report""
Authors' response "the authored decided to adapt the article as a “Proof of concept” manuscript because, considering the exploratory nature of the work carried out."
This manuscript is describing one horse case in the confusing way. The : Proof of concept" does not meet the criteria for an original research manuscript.
Previous comment "L 48-62 and L 83-92 connect and shorten these paragraphs to 6-8 sentences about tarsal ligament desmitis. Remove bold font."
Authors' response "The proposed changes have been introduced."
Both paragraphs (L48-L62 and L83-L92) has not been expressly shortened (currently L48-L60 and L61-L72). Bold font has not been corrected.
Previous comment "L 61 remove Figure 1"
Authors' response "The proposed changes have been introduced."
This comment has been addressed.
Previous comment "L 63-72 An introduction to the anatomy of the ankle ligaments is unnecessary - remove the entire paragraph"
Authors' response "The proposed changes have been introduced."
This comment has been addressed.
Previous comment "L 73-76 The range of motion of the tarsocrural joint is not the aim of the study - remove the entire paragraph"
Authors' response "The proposed changes have been introduced."
This comment has been addressed.
Previous comment "L 93-108 This is not the aim of your study - remove the entire paragraph"
Authors' response "The proposed changes have been introduced."
This comment has been addressed.
Previous comment "L 109- 122 The recently available treatment protocols of tarsal ligament desmitis should be presented briefly."
Authors' response "The proposed changes have been introduced."
This comment has been addressed.
Previous comment "L 123-172 The whole part on regenerative medicine should be shortened to no more than half a page and briefly focused on the strategy of using synovial membrane MSCs enhanced by umbilical cord MSCs"
Authors' response "The proposed changes have been introduced."
This comment has been partially addressed.
Previous comment "L 174 remove Figure 2"
Authors' response "The proposed changes have been introduced."
This comment has been addressed.
Previous comment "L 178 expend abbreviation LMCL. (All abbreviations, including cytokine names (L 185), should be expanded when using them for the first time)."
Authors' response "The proposed changes have been introduced."
This comment has not been addressed. see L72 (CLs), L77 (OA), L81 (mesenchymal stem cells (MSCs) =?, L82 (MSC cell)=?, L89 (Mesenchymal Stem/ Stromal Cells (MSCs)), L 115 (SM-MSCs and UC-MSCs), ...
These types of errors indicate that the manuscript has been prepared carelessly.
Previous comment "L 177-179 I agree with the leading aim of the study. However, the second (L 180-186) can not be supported using the single case report. I recommend that you reword the entire manuscript to focus on the first aim. If this work is to remain a case report, please remove the second aim of the study."
Authors' response "As explained in the answer to the first issue raised by the reviewer, the authors understood the fragility of the article and chose to classify it and rewrite it as a proof of concept."
This comment has not been addressed.
There is no "proof of concept" article type in this Journal. There is Article: Original research manuscripts, Communication, Review, Commentary, Registered Reports, or Case reports. Please, see https://www.mdpi.com/journal/animals/instructions and follow guidelines.
Previous comment "L 221 add the distance between the X-ray tube and the cassette".
Authors' response "The x-ray tube establishes the distance itself with a laser beam so it did not seem to be an important factor for the authors. The laser beam has approximately 1 meter."
This comment has been addressed. It's strange that you don't consider an important parameter of exposure conditions to be important for the repeatability of the study and the evaluation of the results. By the way, the reference distance is 75 cm, and the energy of the radiation beam decreases with the square of the distance.
Previous comment "L 238 If this work is to remain a case report, please move sections 2.5, 2.6, 2.7, 2.8, and 2.9 to supplementary materials or a new article. Continue the report case description with 2.10 and 2.11 subsections (connect seb-subsection 2.10.1 to subsection 2.10)"
Authors' response "This is a “proof of concept” article."
This comment has not been addressed.
There is no "proof of concept" article type in this Journal. There is Article: Original research manuscripts, Communication, Review, Commentary, Registered Reports, or Case reports. Please, see https://www.mdpi.com/journal/animals/instructions and follow guidelines.
Previous comment "L 323-327 I still don't understand the statistical analysis you presented - please provide the raw data you compare (taking into account the number of repetitions and examining the distribution of features). "
Authors' response "As described in the article, the statistical analysis was only carried out to evaluate the concentration of biomarkers present in the secretome of the two cells in comparison, with the samples being evaluated in triplicate. Interleukin concentrations were studied using the ANOVA test, and statistical differences were considered when and significance defined for P<0.05."
This comment has strongly not been addressed.
Previous comment "L 414 The results section should be written to collate detailed clinical and radiological examination results of the horse before treatment and at T3, T4, and T5 so that the progression of changes in each clinical (including AAEP grading) and radiological signs can be tracked."
Authors' response "The proposed changes have been introduced. Radiological examination was not the target of the revaluations as it is described at methods. Ultrasonographic examinations were the focus as well as lameness."
This comment has not been addressed. If yes. I don't see it.
Previous comment "L 437 If this work is to remain a case report, please move sections 3.2 and 3.3 to supplementary materials or a new article."
Authors' response "Please, read the answers above."
This comment has not been addressed.
There is no "proof of concept" article type in this Journal. There is Article: Original research manuscripts, Communication, Review, Commentary, Registered Reports, or Case reports. Please, see https://www.mdpi.com/journal/animals/instructions and follow guidelines.
Previous comment "L 505 The presentation and preparation of the data series for this comparison is completely incomprehensible to me. Please present the raw data, describe how you obtained it what you compare it with, and for what purpose."
Authors' response "The proposed changes have been introduced."
This comment has not been addressed.
Previous comment "L 542 After implementing all the suggested changes, the results obtained might significantly differ from those currently reported in the manuscript. Thus, no specific comments are provided below, and a more thorough review of the discussion section will be provided once the manuscript is considered overall scientifically sound."
Authors' response "The proposed changes have been introduced."
How, if there are no proposed changes to this sentence?
Previous comment "L 705 Conclusions should be thoroughly revised and should be closely related to the purpose of the work. Since it was recommended to limit the purpose of the study only to the description of a clinical case, the entire conclusions section should be rewritten. Unfortunately, there is no room for speculation that is not supported by the research results described in the manuscript."
Authors' response "The proposed changes have been introduced."
This comment has not been addressed.
Unfortunately, in this form, the submitted manuscript does not fulfill the requirements of publication in the Animals journal.
Author Response
Reviewer #2
- Previous comment "The manuscript requires revision following the Instructions for Authors published on https://www.mdpi.com/journal/animals/instructions (...)"
The article has been revised according to the journal instructions.
- Previous comment "Simple Summary sections should appear in all manuscript types". Previous comment "The simple summary is missing".
The reviewer is absolutely right. We apologize for the error. The simple summary has been added. Please check lines 27-38 of the revised manuscript.
- Previous comment "References should be described as follows, depending on the type of work (...)"
"The References section is prepared incorrectly - please carefully check the Instructions for Authors and the method of citing references. This entire section needs improvement. Also, references contained in the manuscript body require improvement and unification - a numerical system is adopted, and references addressed by italics require correction."
We apologize for the inconsistency. All the references were revised and according to the journal instructions the format ACS style was applied in all document through ENDNOTE selection. However, this format does not pre-defined the abbreviation of the journal name.
- Previous comment "Your manuscript is definitely too long and does not fit the set word limit for Case reports. One possible solution to this problem is to transfer some of the information to supplementary materials or to divide this very extensive study into two good, concise works with different goals. One article may actually be a good clinical case (if you follow the guidelines for Case reports) and the other a very good methodological study (if it focuses on cell culture and MSCs preparation protocols). Possibly the latter (MSCs preparation protocols can be entirely transferred to supplementary materials - both from the M&M and results sections). "
This comment has not been addressed. The authors just stated that "the authored decided to adapt the article as a “Proof of concept” manuscript because, considering the exploratory nature of the work carried out". However, this manuscript does not meet the criteria for an original research manuscript.
We apologize, but in this particular case, we do not share the reviewer's opinion. In fact, our article describes a case, but it is a case where a whole new therapeutic strategy was applied and that is exactly what we want to emphasize. This is not a mere description of a clinical case, but rather an entire innovative methodology that was developed and applied to the clinical rehabilitation of an equine. Indeed, the meaning of proof of concept is “to provide evidence, typically deriving from an experiment or pilot project, which demonstrates that a design concept is feasible”. Referring this document to a case report will, due to the word limits imposed, prevent the disclosure of all biotechnology techniques used to obtain the therapeutic outcome achieved. In our point of view, this information included in the materials and methods is essential for the transparency of the clinical research and for the reader's understanding of the therapeutic approach developed. Therefore, moving it to supplementary material will impoverish the manuscript.
According to the journal's rules, “an original research manuscript must report scientifically sound experiments and provides substantial new information”. If the reviewer does not agree, we are available to change it to the format of a “Communication”, where the structure is similar to an article and there is a suggested minimum word count of 2000 words.
Previous comment "The introduction section is too long and does not focus on introducing the topic. The introduction section should be shortened and rewritten according to the detailed comments."
Thank you very much for your comment. The introduction was revised, shortened (much of the previously present text has been eliminated) and rewritten in a more succinct and objective way. Please check lines 58-135 of the revised manuscript.
- Previous comment "The aim of the study requires rewriting."
Thank you very much for your comment. This comment was addressed in order to make the aim of the study clearer. Please check lines 128-135 of the revised manuscript.
- Previous comment "The M&M and results sections require changes described in detailed comments."
We kindly ask the reviewer to take into account the justification given by the authors in point
- Nevertheless, the M&M were reviewed again. The results of clinical examination at the
assessment day were transferred to the results section, now constituting point 3.1. Please
check lines 346-352.
- Previous comment "The discussion section should be strongly restructured and shortened to focus just on the aim of the study. In its current form, no apparent logical order is used in comparing/contrasting the results obtained in the study with previous literature. The authors should attempt structuring the discussion following "the common standard format" which usually consists of the following points:
- One sentence summary that highlights the most relevant results.
- A thorough discussion of each result obtained concerning the corresponding study objective: was the tested hypothesis confirmed or not? Why? What previous evidence supports the specific result or not? It is critical to compare/contrast the result obtained with previous literature in the equine species first, then in veterinary medicine, and finally in human medicine (if not enough data are available for comparison in veterinary medicine
- Statement of study limitations
- Future directions
The conclusions section should be thoroughly revised."
The discussion section was restructured, as suggested by the reviewer. The most relevant results are summarised and discussed in lines 468-491. Study limitations are described in lines 570-576. Future directions are stated in lines 566-567 and 597-599.
Conclusions are stated in lines 584-599.
- Previous comment "L 1 Change "Original Research Manuscript" to "Case report""
Authors' response "the authored decided to adapt the article as a “Proof of concept” manuscript because, considering the exploratory nature of the work carried out."
This manuscript is describing one horse case in the confusing way. The “Proof of concept" does not meet the criteria for an original research manuscript.
We kindly ask the reviewer to take into account the justification given by the authors in 4.
- Previous comment "L 48-62 and L 83-92 connect and shorten these paragraphs to 6-8 sentences about tarsal ligament desmitis. Remove bold font."
Authors' response "The proposed changes have been introduced."
Both paragraphs (L48-L62 and L83-L92) has not been expressly shortened (currently L48-L60 and L61-L72). Bold font has not been corrected.
As proposed in 5., the Introduction section has been completely revised.
- Previous comment "L 123-172 The whole part on regenerative medicine should be shortened to no more than half a page and briefly focused on the strategy of using synovial membrane MSCs enhanced by umbilical cord MSCs"
Authors' response "The proposed changes have been introduced."
This comment has been partially addressed.
As proposed by the reviewer in points 5. and 10, the Introduction section has been completely revised.
- Previous comment “L 178 expend abbreviation LMCL. (All abbreviations, including cytokine names (L 185), should be expanded when using them for the first time).”
Authors’ response “The proposed changes have been introduced.”
This comment has not been addressed. See L72 (CLs), L77 (OA), L81 (mesenchymal stem cells (MSCs) =?, L82 (MSC cell)=?, L89 (Mesenchymal Stem/ Stromal Cells (MSCs)), L 115 (SM-MSCs and UC-MSCs), …
These types of errors indicate that the manuscript has been prepared carelessly.
We apologize for the errors. We believe that this time everything is corrected.
- Previous comment "L 177-179 I agree with the leading aim of the study. However, the second (L 180-186) can not be supported using the single case report. I recommend that you reword the entire manuscript to focus on the first aim. If this work is to remain a case report, please remove the second aim of the study."
Authors' response "As explained in the answer to the first issue raised by the reviewer, the authors understood the fragility of the article and chose to classify it and rewrite it as a proof of concept."
This comment has not been addressed.
There is no "proof of concept" article type in this Journal. There is Article: Original research manuscripts, Communication, Review, Commentary, Registered Reports, or Case reports. Please, see https://www.mdpi.com/journal/animals/instructions and follow guidelines.
We kindly ask the reviewer to take into account the justification given by the authors in 4. As proposed in 5., the Introduction section has been completely revised and the aims of the study were also clarified. Please check lines 130-137 of the revised manuscript.
- Previous comment "L 221 add the distance between the X-ray tube and the cassette".
Authors' response "The x-ray tube establishes the distance itself with a laser beam so it did not seem to be an important factor for the authors. The laser beam has approximately 1 meter."
This comment has been addressed. It's strange that you don't consider an important parameter of exposure conditions to be important for the repeatability of the study and the evaluation of the results. By the way, the reference distance is 75 cm, and the energy of the radiation beam decreases with the square of the distance.
Many thanks for the comment. In fact, it is agreed that this factor is of extreme importance and that’s why the manufacturer and brand mark of the x-ray is presented, as well as the constants used. With this x-ray tube, the beam distance is always the same, regardless of the user. Nevertheless, the specification requested was corrected in accordance with the manufacturer specifications.
- Previous comment "L 238 If this work is to remain a case report, please move sections 2.5, 2.6, 2.7, 2.8, and 2.9 to supplementary materials or a new article. Continue the report case description with 2.10 and 2.11 subsections (connect sub-subsection 2.10.1 to subsection 2.10)"
Authors' response "This is a “proof of concept” article."
This comment has not been addressed.
There is no "proof of concept" article type in this Journal. There is Article: Original research manuscripts, Communication, Review, Commentary, Registered Reports, or Case reports. Please, see https://www.mdpi.com/journal/animals/instructions and follow guidelines.
We kindly ask the reviewer to take into account the justification given by the authors in 4.
- Previous comment "L 323-327 I still don't understand the statistical analysis you presented - please provide the raw data you compare (taking into account the number of repetitions and examining the distribution of features). "
Authors' response "As described in the article, the statistical analysis was only carried out to evaluate the concentration of biomarkers present in the secretome of the two cells in comparison, with the samples being evaluated in triplicate. Interleukin concentrations were studied using the ANOVA test, and statistical differences were considered when and significance defined for P<0.05."
This comment has strongly not been addressed.
In fact, the reviewer has a point. The authors realized that the statistical analysis of interleukin concentrations in both synovial and umbilical cells did not bring any advantages or novelty to the results. Indeed, the secretome of cells in vivo acquire different behaviour depending on the environment in which they are inserted. Therefore, the data regarding the synovial cells were eliminated from this document as they were already reported in another manuscript of our research group. This way, raw data regarding UC-MSCs is presented in supplementary materials.
- Previous comment "L 414 The results section should be written to collate detailed clinical and radiological examination results of the horse before treatment and at T3, T4, and T5 so that the progression of changes in each clinical (including AAEP grading) and radiological signs can be tracked."
Authors' response "The proposed changes have been introduced. Radiological examination was not the target of the revaluations as it is described at methods. Ultrasonographic examinations were the focus as well as lameness."
This comment has not been addressed. If yes. I don't see it.
Clinical changes and lameness grading are present in the section - “Treatment results”.
As mentioned before, radiological examination was only performed at the initial clinical examination and was not repeated. Only clinical and ultrasonographic examinations were performed in the follow-up period, as radiological examinations are not so specific for ligament injuries. This is clearly mentioned at line 360.
If the reviewer finds it pertinent, we can take the previous radiological examination off the work. However, this technique was important in the clinical assessment to discard the other tarsal lesions.
- Previous comment "L 437 If this work is to remain a case report, please move sections 3.2 and 3.3 to supplementary materials or a new article."
Authors' response "Please, read the answers above."
This comment has not been addressed.
There is no "proof of concept" article type in this Journal. There is Article: Original research manuscripts, Communication, Review, Commentary, Registered Reports, or Case reports. Please, see https://www.mdpi.com/journal/animals/instructions and follow guidelines.
We kindly ask the reviewer to take into account the justification given by the authors in 4.
- Previous comment "L 505 The presentation and preparation of the data series for this comparison is completely incomprehensible to me. Please present the raw data, describe how you obtained it what you compare it with, and for what purpose."
Authors' response "The proposed changes have been introduced."
This comment has not been addressed.
Please take into consideration the answer to the reviewer’s comment in point 16. With this regard, the results obtained were clarified in lines 430-432 of the revised manuscript.
- Previous comment "L 705 Conclusions should be thoroughly revised and should be closely related to the purpose of the work. Since it was recommended to limit the purpose of the study only to the description of a clinical case, the entire conclusions section should be rewritten. Unfortunately, there is no room for speculation that is not supported by the research results described in the manuscript."
Authors' response "The proposed changes have been introduced."
This comment has not been addressed.
We kindly ask the reviewer to take into account the justification given by the authors in 4.

Round 3
Reviewer 2 Report (Previous Reviewer 3)
Comments and Suggestions for Authors
Dear Author,
Thank you for addressing the comments. The most important errors have been corrected and the manuscript can be accepted for publication.
Good luck with your further work.
This manuscript is a resubmission of an earlier submission. The following is a list of the peer review reports and author responses from that submission.
Round 1
Reviewer 1 Report
Comments and Suggestions for Authors
I carefully read the manuscript titled “Treatment of Equine Tarsus Long Medial Collateral Desmitis with allogenic synovial membrane mesenchymal stem cells enhanced by umbilical cord mesenchymal stem cell-derived conditioned medium: case report” from Reis et al. Authors stated having being submitted a “Case Report” manuscript regarding treatment of desmitis of tarsus of long medial collateral ligament, occurred in a seven years old sporty horse. Treatment in object is a combination of expanded MSC from donor synovia and a concentrated culture media conditioned by donor MSC derived from Warton’s Jelly.
In the Introduction section Authors included an extended (lines 47-249), presentation of the state-of-the-art in the field of horse ligament injuries including; anatomy, pathogenesis, diagnosis and diagnostic procedures as well as possible treatments not only regarding cell-therapies. Nonetheless, the description of the clinical case is limited to ten lines from line 268 to 278. Subject description and clinical evaluation are then fragmentarily reported in Methods section, unnecessarily divided into many paragraphs and sub-paragraphs (lines 289-334). Similarly method regarding cells and CM preparation, treatment, rehab program and post-treatment monitoring resulted too much extensively described and divided in a number of sub-paragraphs. Thus, my major comment regards the manuscript preparation, as I did not find any conceptual or logical errors.
The manuscript has not been prepared as a Case Report:
- Introduction has been drafted with the intent to describe the knowledge in the field (from basic knowledge to advanced hypothesis) instead of presenting the clinical case, resulting in a long and unnecessarily detailed section citing more than 40 references. This section should be strongly shortened focusing on the presentation of the case and condensing the clinical evaluation and diagnostic exams in the same section. Consequently figure 1 and 2 are not necessary.
- -Methods should be strongly revised limiting to the description of the methods to obtain cells and CM (described briefly), protocol of treatment administration and rehab program. In paragraphs 2.2 (lines292-299) Authors describe the design of a prospect study and the inclusion criteria, but this is a case report, thus the paragraph should be removed.
- In the Result section the whole 3.1 paragraph should be removed and data should be included and briefly described in the clinical evaluation in the Introduction section as explained above. Conversely, paragraph 3.2 resulted poorly written; this section should presented analytical data as quality control (QC) of the cell product obtained. QC of the manufactured cell product is one of the most important issue to address in cell-therapy and could not be condensed in simple statement as “SM-MSCs were successfully isolated and expanded from the donor” or simply citing papers the applied the same protocol. Each cell preparation should pass the QC test, also if the manufacturing protocol applied is the same. Even if the procedure applied an of-the-shelf preparation, where the cells are already characterized, QC test is however mandatory at the time of thawing and administration. Typical QC test reported number of cells, vitality, immunophenotype, differentiation potential, etc. Similarly, paragraph 3.3 should present data on the CM to confirm the quality of the preparation. The comparison of the two MSC sources, in terms of production of biomolecules, is out of topic and goes beyond the aim of this clinical case report. Moreover, the demonstration of the superior performance of UC-MSC should be supported by more rigorous data (comparison among a number of different batches for instance), I strongly suggest limiting the description to the biomolecules concentration in the CM applied at this specific clinical case.
- Page 16 line 565: the statement “there was a complete regeneration of the ligament” is not supported by data. Even if good ultrasonographic images could suggest a clinical recovery the demonstration of tissue regeneration should be supported by histological analysis, which is not applicable. Thus, I suggest avoiding any reference to the regeneration of the ligament tissue (also at line 754) while stressing the clinical and sporting recovery as well as the absence of relapse during the observation period. In this view, I found those important information only at figure 9 legend (lines 578-581), while deserving to be mentioned in the main text.
- Discussion should be completely re-edited. In its actual form it represents an unnecessarily long revision of the literature in style of a Review paper. Discussion should be focused on the specific clinical experience, what it could suggest or how this clinical observation could trigger future studies. As Case report description could be open to interpretation lacking any empirical demonstration, discussion on limits and criticism of the treatment applied would be appreciated.
Concluding, I strongly suggest revising the journal’s instruction for Authors and re-edit the manuscript according to the type of publication, also taking in account the word count limit.
In particular, here the guidelines extracted for the “Animals” instructions:
“Case reports: must provide an in depth, rather than superficial, review of a particular case. The purpose of this presentation should be that it suggests a novel way of interpreting existing knowledge on the topic. Articles are usually identified as case studies rather than scientific papers if they contain material that is deemed to be open to interpretation, rather than empirically proven. Case studies, like Commentaries, must contain reasoned arguments, not just arguments expressing the emotions of the author. Case studies must be concise, well argued, and erudite, as well as be written to a very high standard of English. All submissions will be submitted for peer review. Case reports should have a main text of around 2500 words.”
Minors:
-Terminology for MSCs should be revised. ISCT defined these cells as multipotent mesenchymal stromal cells maintaining the acronymous MSCs [Horwitz EM, et al (2005) Clarification of the nomenclature for MSC: The International Society for Cellular Therapy position statement. Cytotherapy. 7(5):393-5] and avoiding any reference to the stemness of these cells. The scientific community generally uses this nomenclature, these last years. Entire manuscript needs nomenclature revision when referred to “mesenchymal stem cells”.
-Title: “…Equine Tarsus Long Medial Collateral Desmitis…” should be “…Equine Tarsus Long Medial Collateral Ligament Desmitis…”
- Typo at: line 637, line 661
Comments on the Quality of English Language
Language revision from a natural speaker is needed
Reviewer 2 Report
Comments and Suggestions for Authors
Dear authors,
thank you very much for sending me this interesting manuscript. I would like to comment directly on the following points:
1) Overall, I see here a mixture of two works that should be published separately. On the one hand the outcome of the Secretome and on the other hand the actual case report. Then some points would be emphasised more clearly and there would be no confusion between statistically evaluable results and the case report itself (n=1).
2) MSC are also called medicinal signaling cells (see https://doi.org/10.1002/sctm.17-0051) for the reasons given. I would include this in the manuscript.
3) The manuscript needs to be checked again for correct grammar. Also, many double blanks, missing spaces, etc. have been noticed.
4) In S.I. units, there is always a space between unit and value, this also needs to be revised throughout the manuscript. In addition, commas should be placed in the same place instead of dots for numerical values.
5) In the methodology section, the order numbers for media and supplements should be given. 3% Pen/Strep may be misleading.
6.) line 326: Please indicate that it is a linear transducer. Here too: Comma instead of dot for the MHz specification
7) line 336: Please state the exact age of the foal (a few days, 2 weeks?).
8) line 353: From how many umbilical cords was the tissue taken, was an animal experiment application necessary for this (even if it is a "waste product"), was it from the same foal as the other cells were obtained?
9) line 364: Typo: DMEM-LG (not DMED-LG)
10) line 384 and lines 414, 422: RPM must be replaced by a g-number, otherwise it is not possible to guess with which force centrifugation was performed.
11) line 413: What are "dry blood collection tubes"? Do you mean serum collection tubes? Then I would also write it that way. These also have an activator so that the blood can coagulate. I would include this (manufacturer is sufficient).
12) line 417: Filtering was not done with a syringe, but with a syringe filter with 0.22 µm.
13) line 426: 10x10^6. Technically, then, you have to write 1x10^7.
14) line 432: There is a dash in front of LMCL and a minus behind it. This should also be gone through again in the manuscript.
15) Line 443: Please indicate the size of the needle for injection.
16) Line 506/507: The symbols in the brackets are missing here.
17) Fig. 8 and Tab. 3: These look great, but either belong in a separate publication as noted at the beginning or need to be significantly improved: Statistical methods are missing, how large is the n-number for this, etc. Please improve! This also applies to the data in the table, what does the SEM (n-number) refer to?
18) line 637: Typos of IL-6 and dueto (missing spaces).
19) Italicised words appear in various places in the manuscript. E.g. line 422 "1600 rpm". Please also check the entire manuscript.
20) line 866, references: Why are the volume numbers in bold?
21) reference 21: Authors in all capital letters.
22) line 958: reference 56: An enter is missing here.
23) Lastly, on the following parts of the discussion concerning the specific case report: lines 748-750, 753 - 754 and 757 - 759. For a case report, one can only say that the therapy was more successful than expected and that this may have been promoted by the treatment (i.e. it was "successful"). Whether this is due to the eSM-MSCs or the CM of the eUC-MSCs or their patented combination cannot be determined at this point as written here. Please make this more clearly visible in the manuscript. As described in the nature of medicinal signalling cells, in a favourable case, the injection alone can reactivate the patient's local immune system by reactivating it in the joint, and thus also lead to a desired result.
In summary, the latter point and the statistics question result in a major revision. I hope to have helped you with this and look forward to your reply.
Comments on the Quality of English Language3) The manuscript needs to be checked again for correct grammar. Also, many double blanks, missing spaces, etc. have been noticed.
Reviewer 3 Report
Comments and Suggestions for Authors
The work is interesting, however, the way it is presented requires a thorough revision and concentration either on the proper description of the case report or the proper description of the research. You can't mix a little of this and a little of that. Especially by presenting statistics that are not substantiated. L 541-548 This presentation of numerical data is unacceptable. You describe the case of ONE HORSE and present statistics for interleukin concentrations. How? There is no mention of comparing anything in your methodology at all, and certainly not concerning a single case.
I regret to say that the entire chapter 1 (L 46 - 279) is about everything and nothing. The introduction section should be a coherent introduction to the topic covered by the research in a maximum of 1 - 1.5 pages. In this manuscript, the introduction is far too long and contains several pieces of information that are not relevant to the study's purpose. On the other hand, pieces of information that are important to provide context to the reader are not thoroughly reported. This section, like most sections in this manuscript, requires extensive rewriting.
The purpose of the work is missing at the end of the introduction section.
The simple summary is missing.
Are these figures your own? If they do not, the proper reference support is required.
How figures are cited in the text requires improvement.
All abbreviations, including cytokine names, should be expanded when using them for the first time.
The References section is prepared incorrectly - please carefully check the Instructions for Authors and the method of citing references. This entire section needs improvement. Also, references contained in the manuscript body require improvement and unification - a numerical system is adopted, and references addressed by italics require correction.
I regret to say that the article was prepared unprofessionally and carelessly. Authors should read the latest case reports published in the Animals Journal and improve their work by following their example. Then the revised version of this manuscript can be assessed again after resubmission.